# Approaches to Health Efficiency across the European Space through the Lens of the Health Budget Effort

**DOI:** 10.3390/ijerph19053063

**Published:** 2022-03-05

**Authors:** Valentin Marian Antohi, Romeo Victor Ionescu, Monica Laura Zlati, Cristian Mirica, Nicoleta Cristache

**Affiliations:** 1Department of Business Administration, Dunarea de Jos University, 800008 Galati, Romania or monica.zlati@ugal.ro (M.L.Z.); cristian.mirica@ugal.ro (C.M.); cristache.nicoleta@yahoo.de (N.C.); 2Departament of Finance, Accounting and Economic Theory, Transylvania University, 500036 Brasov, Romania; 3Department of Administrative Sciences and Regional Studies, Dunarea de Jos University, 800201 Galati, Romania; ionescu_v_romeo@yahoo.com; 4Department of Accounting, Audit and Finance, Stefan cel Mare University, 720229 Suceava, Romania

**Keywords:** efficiency of budget allocation, health, pandemic, sanitary vulnerability, social protection

## Abstract

In the context of the COVID-19 pandemic, financial resources allocated to the health system have been refocused according to priority 0: fighting the pandemic. The main objective of this research is to identify the vulnerabilities affecting the health budget effort in the EU and in the Member States during the health crisis period. The analysis takes into account relevant statistical indicators both in terms of financial allocation to health and expenditure on health protection of the population in the Member States, with the effect being tracked even during the pandemic period. The novelty of the study is the identification of viable directions of intervention based on the structural determination of expenditures related to measures to combat the pandemic and making proposals for changes in public policies based on the determination of the effectiveness of budget allocations in health in relation to the proposed purpose. The main outcome of the study is the identification of the vulnerabilities and the projection of measures to mitigate them in the medium and long term.

## 1. Introduction

Efficient financing of health services in the European area is a priority from the perspective of sustainable development objectives as well as from the perspective of the European social protection and health policies. From the financial distribution point of view, in line with the European policies, there is more monitoring of the financial indicators especially in terms of improving the health of the population in the long term in relation to increasing the quality of the health services, attracting specialists in the medical system or improving the quality of life in relation to new treatments and medical protocols in the field.

Thus, according to the ISQua (International Society for Quality in Health Care) guidelines on quality in public health services, the main objectives pursued in terms of ensuring the health of the population are aimed at forming centres of performance, ensuring patient safety, continuously improving the quality of health services, implementing new technologies in health and ensuring quality and safety through the sustainable use of resources [1].

These issues have led to the creation of international service quality standards, which are implemented differently at the EU level (for example, in Romania the second edition of the revised quality assurance standards for health services is implemented at this moment) [2]. This brings with it disparities in resource consumption and population health insurance; all the more, in the context of COVID, vulnerabilities have generated crisis situations in countries that have faced certain risk factors triggering health events during the pandemic (Italy, Spain, France, Sweden, Romania).

In view of the above, the aim of the present research is to identify the vulnerabilities affecting the health budget effort during the health crisis.

Thus, we define the following research objectives:

**O1:** Identify the pre-pandemic context of the health of the European population in relation to the financial allocations for health and the level of social protection offered by the Member State governments to their citizens.

**O2:** Quantify the level of damage to the health profile of the population following the outbreak of the pandemic.

**O3:** Determination of the existence of a regression correlation between annual health status and morbidity and mortality rates induced by the COVID-19 pandemic.

**O4:** Identify measures to address vulnerabilities arising from the misallocation of funds in health, based on observations made during the study.

Results of the study have led to relevant and useful conclusions for the financial and governmental decision-makers in order to improve the efficiency of the budget allocations, with a direct effect on improving the long-term health status of the population.

Further research covers the following sections: Literature Review, Methodology, Results and Discussion and Conclusions. The literature review provides a conceptual framework for defining our proposed model. The methodology highlights the manner of scientific research capable of supporting the definition and implementation of the proposed model, while Results and Discussion provides the necessary framework for national and supranational decision-makers in the context of a new approach to health financing. The conclusions are the quintessence of the present research, highlighting the novelty of the study, its limits, its usefulness and possible further developments.

## 2. Literature Review

The present study covers an area of interest regarding financing of the health services, which was widely debated in the literature even before the outbreak of the pandemic [3]. Between 2000 and 2018, more than 70,000 articles on public health financing were written by international authors. Compared to the period 2000–2018, in the period after the outbreak of the pandemic, there has been an intensification of the research effort; in only 3 years, more than 45,000 articles have been published on this topic (Google Academics, 2021).

Using the VOSviewer software (version 1.6.18, developed by Centre for Science and Technology Studies, Leiden University, The Netherlands ), a total of 74 articles published in 2020 and 2021 on the research topic were reviewed (see Figure 1 and the Appendix A). A clustering of research interest has been identified in 3 clusters, linking the health system financing issues under pandemic crisis (green cluster), public health policies and the main social effects of the pandemic (blue cluster) and repercussions on economic life (depression, anxiety and mental status of the population—red cluster). This has led to deepening unemployment, economic recession and deterioration of the health status of the population with an impact on the health system.

The main themes addressed by the specialists focused on the challenges of the pandemic in relation to the public health system, the financial crisis induced by the pandemic and systemic difficulties faced by some countries in dealing with the pandemic, as well as the vulnerabilities of the public expenditure financing system following the restructuring of the expenditure in line with the immediate measures required to combat the pandemic (Figure 2).

Following the literature review, we noted the concern of some authors [4] regarding the financial allocation for health services in relation to the minimum set of social services to support an adequate state of health of all citizens. Through contributory mechanisms, the authors identify the main categories of contributors in the public and private sector in order to establish sufficient financial sources to ensure a minimum necessary universal protection against the low health status of the population at risk of poverty and to promote an equitable system of access to health services. Given the characteristics, benefits and advantage/disadvantage system of publicly and privately funded services, the authors conclude that the future will see mixed models of health financing support in EU Member States different from the current ones.

In the same context of transparent and sustainable financing of health systems, some authors [5,6,7,8,9,10] address the financial structure of public health expenditure in a dynamic way through studies addressed to European countries. In Eastern European countries [6], a higher need for health system financing is observed based on the dynamic analysis carried out in the period 2000–2017, which reveals that public expenditure has been decreasing in most countries (the maximum reduction being reached by Bulgaria and the Czech Republic), while government taxation for social insurance had an increasing trend in most countries. The Bulgarian system is surprising, whose financing structure seems to have shifted toward social health insurance at the expense of public spending or income taxation with a percentage contribution to the health insurance.

According to some authors [11], the pandemic has combined two different types of crisis—health and economic—which together have affected national health systems, and budgetary rebalancing measures are recommended, corroborated by an increase in the degree of collection and an increase in the capacity of the system to generate tax payments, which is the measure of sustainable development of an economic system. The same problem-solution approach is found in their research [12], which emphasizes the need for fiscal decentralization, reforming the system so that the interaction between economic agents and control bodies involves a novel approach to the taxpayer-state representative relationship.

During the pandemic, economies of the political powers were affected, in particular due to the restriction of economic trade, so that the governments of these countries had to act in terms of health protection but also to stimulate the affected domestic business environment in the global trade segment in order to recover. The authors [13] propose a system based on PPML (Poisson Pseudo-Maximum Likelihood) estimators, so that the financial development mechanism strengthens public health care by promoting financial flows through state-owned firms. While this may be negative in the long term, in the short term, the proposed solution may achieve its goal.

In another approach, [14] consider the pandemic crisis as a challenge to identify measures with maximum efficiency and effectiveness, and some innovation solutions in health through public subsidies are proposed, identifying 23 opportunities that could constitute solutions in the segment of public policies, telemedicine, testing and diagnostics, social science, etc.

Iizuka [15] provides a brief analysis of the impact of the pandemic on the health of the population and the economy as a whole. Based on the existing disparities between national health systems, countries around the world faced a wide range of problems during the pandemic, and the author concludes that there is a need for ex-ante monitoring and a concomitant mobilization of the private and public sector for the sustainability of the implementation of measures to combat effects of the pandemic crisis.

The challenge of making health systems resilient to shocks and crises can be taken up in their view [16] through a sustainable transformation of system financing to ensure sufficient resources for prevention and control, surveillance and effective information through digitization. The contribution of telemedicine and the mobilization of the private sector are not ignored; they contribute to the implementation of viable solutions, multisectoral cooperation being the basis for the sustainable resilience of the system.

Another approach that points out critical aspects of some health systems, such as the American one, brings to the fore the fragmentation of the health care system and inadequate social protection, which in pandemics can be major obstacles and threats to the health of the population [17].

An analysis by [18] highlights the major impact of COVID-19 on the financial system as a whole, considering that two factors are disruptive in any forecast: the decrease in wage, tax and VAT receipts by 9%, correlated with the historical progression of expenditure (€23.3bn) in the French health system. This development will favour the social security deficit assessed for the 2023 horizon, which includes health and risk insurance.

Ref. [19] points out that the pandemic has induced a global state of vulnerability to the medical phenomenon, with the countries of the world unprepared for what followed. The management of the pandemic resulted in lockdowns in most countries, which led to restrictions and halts in social and economic activities. The solutions proposed by the authors are pertinent and refer to improving the performance of the hospital superstructure, setting up a disease monitoring interface, access to advanced health technologies and health education of the population. These measures need to be financed in a quasi-unitary way by the countries of the world, which will lead to changes in health financing policies.

An analysis of health system financing in Nigeria in response to the outbreak of the pandemic is by [20], which shows that for developing countries, the financial strain was and is burdensome and required the establishment of stringent fiscal policies and health insurance packages that contributed to fiscal pressure.

The authors [21] show that the pandemic created financial market disruptions due to underestimation of health risk and slow response to financial stress during the period. These elements can be combined in the future for the creation of behavioural profiles during the crisis, the analysis being carried out on the 23 largest countries in the world with the exception of Russia and China.

Other authors consider that the health system response in Arab countries is more efficient than in other countries [22], arguing for a 3-step algorithm containing detection, prevention and containment and treatment. In this sense, public health policy in Arab countries has been based on testing, social distancing and monitoring of infected people to a greater extent than in other countries. In addition, the population showed a better understanding of social distancing measures and travel restrictions. Third, economic support measures contributed to the efficiency of the Arab health-care system during the pandemic.

An X-ray of the EU health system in a pandemic context by [23] shows that there is a big difference in the impact of the pandemic in the Member States, both in the severity of the impact (number of deaths due to the pandemic) and in the financial pressure on public health systems. The only countries where public pressure on the health system is not of a financial nature are Sweden, Germany and Luxembourg, the rest being affected by the health security measures that have been implemented.

In a dynamic approach, [24] compared the structure of health systems and the financial effects of the pandemic in countries with mixed (public-private) health insurance systems such as the US, UK, Germany and Israel. The negative effects on health have been most severe in the US, where the prevalence of activity-based payment system and direct government control has been manifested to a limited extent. The authors show that during the pandemic period the pressure fell on the public health system to which low-income people turned.

The public health system is considered the basis of a competitive and sustainable health system because it solves community problems and promotes social equity in health in their opinion [25]. The authors show that global cooperation has contributed to better planning and urgency in the implementation of solutions, as well as to the development of viable strategies to combat viruses based on the principle of accumulation of experience. From a financial point of view, the financing capacity of health systems must ensure greater flexibility to respond to exceptional emergencies. Telemedicine and remote triage are brought into discussion.

A systematic review of the literature by [26] reflects the fact that, in the case of Eastern European countries (Romania), the main threats affecting the national health system are the inadequate health risk factor behavior of the population and the decrease in the number of specialists working in the health care, on the one hand, and on the other, the high incidence of contagious and chronic diseases and the poor living conditions of the majority of the population, especially in rural areas.

A model of health system differentiation based on the identification of the economic inequalities has been addressed in several studies [27,28,29]. In the paper Differentiation in Healthcare Financing in EU Countries [27], the model for assessing economic inequality is presented by referring to the degree of concentration of the distributions which shows that there are major inequalities between the financing of health services over the period 2013–2017, a period in which Romania ranked last in terms of the share of health expenditure in GDP. The disproportionalities identified by the disturbance indicator show that each Member State needs to bring its health policies closer to the economic condition and to correlate the health expenditure per individual with the objectives for the level of affordable health.

A prospective study on the dynamics of the health services in Eastern European countries shows that, in terms of health expenditure per capita, Romania ranks last, but ahead of Croatia and Slovenia [30]. Thus, the authors opine that constant monitoring of economic trends and the health status of the population is required, and the absence of such health screening may constitute a source of substantial public expenditure on health (by deteriorating the health status of the population).

According to other authors [31,32] there are several clusters in the EU formed on the basis of stratification of the level of health security of the population, European inhomogeneity being a problem in terms of unitary implementation of public health policies, health management and technological uniformity at the European level. Thus, there is a need for rebalancing of the cyclical assessments at a national level.

During the pandemic period, it was found that there was an impairment of the economic and medical performance indicators [33], mainly due to the impairment of fixed expenses on the background of the reduction in the number of patients and the partial relocation of hospital activities to the COVID-19 area. The disease control measures have also affected the health budget balance. In the study conducted by the authors [33], it is shown that through the DEA-BCC model (Data Envelopment Analysis-Banker, Charnes and Cooper Model) in Romania a number of 18 COVID-19 support hospitals were evaluated in the period 2019–2020, of which only 5 were identified as efficient units. The other 13 ranked below the efficiency indicators in terms of quality indicators, sufficient nursing staff, sufficient number of beds and health outputs delivered by hospital units.

According to the report for Romania [34], it is socio-economically and demographically below the European indicators, with a poverty rate of 23.6% (compared to the EU average of 16.9%) and a GDP/capita of 18,800 Euro PPP (Purchasing power parity), (compared to the EU average of 30,000 Euro PPP). In terms of health status, in line with the general European trend, Romania has been on a positive trend during 2007–2013, with an increase in life expectancy from 71.2 years in 2000 to 75.3 years in 2017, which is below the European limit for the EU (80.9 years). The main factors behind this development are educational attainment, incidence of heart disease and cancer mortality. The OECD report shows that the health system in Romania is the poorest among the European countries in terms of expenditure per capita (Romania spends 1030 euros compared to 2884 euros in the EU), but also in terms of size of allocation of GDP (5% compared to 9.8% in the EU).

In terms of ways to combat the pandemic, researchers [35] have focused on studying the most effective method (vaccination), conducting a study of the effects of vaccination and the general framework for implementing security measures, a framework that resides in 3 distinct elements; namely, security of the vaccination solution (limiting adverse effects and uncertainties of the process), funding to ensure equal opportunity for vaccination and ensuring adequate ethics of the process both in terms of the vaccination action and in terms of recognition of vaccination and use of information during the monitoring period for continuous improvement of the vaccination solution.

In his opinion [36], the COVID-19 experience brings to the attention of global decision-makers the improvement of the health framework, the strengthening of national public health systems, the identification of health risks and the increase of awareness of these risks, so that events of pandemic magnitude can be better addressed and more efficiently managed.

These results published in the literature lead to the conclusion that there is a need to reposition funding in relation to the actual need for funding, especially in light of the pandemic that has influenced and continues to influence the health of the population and the effects of funding health services during the pandemic.

In order to achieve the purpose of the research (to identify the influencing factors on the efficiency of budget allocation in health), we defined the following working hypotheses:

**H1:** 
*The health status of the population varies in direct proportion to the health budget allocations, and changes in the government policies are likely to have a direct proportional influence on the medium and long-term health balance.*


**H2:** 
*In the case of financial policy stability for health, the main deflator of the allocation efficiency is the social protection expenditure as a percentage of GDP.*


**H3:** 
*In the case of health financial policy stability, the pandemic morbidity rate tends to increase in magnitude, with the policy update level being a maximum of 10 years.*


**H4:** 
*Under conditions of the health promotion through financial allocations for social protection or for strengthening the health security of the population, the vaccination rate tends to decrease if the funded level of the social security is not sustained over time.*


## 3. Methodology

We collected over a statistical period of 11 years information on the dynamics of economic and financial indicators representative for health system financing (EUROSTAT, 2020), correlated with the influence of morbidity and mortality indicators due to the COVID-19 pandemic. These data were then concatenated into a consolidated database on which statistical procedures were applied to obtain regression and prediction equations using IBM SPSS 25 statistical software.

Thus, we used the Eurostat database [37], from which we collected information on the evolution of the following indicators at Member State level for the period 2009–2020. In addition to the Eurostat indicators, we analysed the information provided by Google News [38] on morbidity and mortality rates related to the COVID-19 pandemic for each EU27 Member State at the beginning of November 2021 (see Table 1).

Collected information allowed for the creation of an integrated database which was modeled using the multiple linear regression process by the mathematical method of least squares. Annual models of the correlation between total health care expenditure by EU27 average expenditure dependent variable and the regressors COVID_VACC, COVID_ILL, *SPR_ADM_year_*, *SPR_REC_SUMT_ESC_year_*, *SPR_OTHER_year_*, *SPR_REC_SUMT_GOV_year_*, *HLTH_SHA11_LT*_*year*−1_, *HLTH_SHA11_HC*_*year*−1_, *SPR_EXP_SUM_year_*, *HLTH_SHA11_LT_year_* were obtained.

The general model is presented as the following regression equation:(1)$$\begin{array}{cc} {HLTH}_{{SHA11}_{{HC}_{year}}} & =\alpha_{1}*\;{SPR}_{{ADM}_{year}}\\ & +\alpha_{2}*\;{SPR}_{{REC}_{{SUMT}_{{ESC}_{year}}}}\\ & +\alpha_{3}*\;{SPR}_{{OTHER}_{year}}+\alpha_{4}*\;{SPR}_{{REC}_{{SUMT}_{{GOV}_{year}}}}\\ & +\alpha_{5}*{HLTH}_{{SHA11}_{{LT}_{year-1}}}+\alpha_{6}*\;{HLTH}_{{SHA11}_{{HC}_{year-1}}}\\ & +\alpha_{7}*{SPR}_{{EXP}_{{SUM}_{year}}}+\alpha_{8}*{HLTH}_{{SHA11}_{{LT}_{year}}}\\ & +\alpha_{9}*{COVID}_{VACC}+\alpha_{10}*{COVID}_{ILL} \end{array}$$
where: $HLTH\_SHA11\_HC_{year}$—dependent variable of the model in year n; n ∈ [2009, 2020]; $\alpha_{i}$—coefficients of the regression variables, i ∈[1,10]; COVID_VACC, COVID_ILL, *SPR_ADM_year_*, *SPR_REC_SUMT_ESC_year_*, *SPR_OTHER_year_*, *SPR_REC_SUMT_GOV_year_*, *HLTH_SHA11_LT*_*year*−1_, *HLTH_SHA11_HC*_*year*−1_, *SPR_EXP_SUM_year_*, *HLTH_SHA11_LT_year_*—regressors of the model in year n.

## 4. Results and Discussion

We developed annual models based on the general model for which we obtained the following tests of statistical significance (see Table 2):

It is observed that the statistical significance of the overall model is above 99%, which means that the variables are homogeneous and allow for quantifying the efficiency function of the financial allocation in health in relation to the proposed regressors.

In terms of change statistics, the F-test shows a maximum value in 2015, a year in which we can state that the efficiency of the financial allocations in relation to the objectives of quality, security and social resilience was maximum. The statistical representation function is maximal. This was also the case in 2018, when the F-test value was the second highest in the annual series.

It is noted that the year 2020 showed major disturbances in the F-test, with the F-test decreasing more than 5 times compared to 2015 and more than 4 times compared to 2018. As a result, the efficiency of the health allocations during the pandemic period was affected by the social protection measures and the disease control measures, which affects the health balance of the system.

From the Durbin–Watson function point of view, the results are of high homogeneity and statistical representativeness, the value of the test being close to 2, which allows validating the data series as homogeneous and representative for the studied phenomenon.

The results of the ANOVA test reflect that the residual quantity of the regression function is insignificant relative to the sum of the regression squares, with a representation of 10 degrees of freedom out of a total of 26 possible. The regression function F shows a value of up to 2266 points in 2015 and 443 in 2020 and a Sig coefficient tending to 0 (see Table 3).

In Table 3, it can be seen that the sum of the residual squares of the annual patterns decreases during the period under analysis, both in terms of quantity and as a share of the total sum of squares. This aspect demonstrates that the decadal policy update period in the health sector shows reflexivity in relation to the predicted output of the dependent variable $HLTH\_SHA11\_HC_{year}$. The test results confirm hypothesis H3; namely, in the case of health financial policy stability, the pandemic morbidity rate tends to increase in magnitude, with the policy update level being a maximum of 10 years. Our approach is also supported by [28,34].

Pearson correlation coefficients for the dependent variable in relation to the change in the dependent variable from the previous year and the analyzed regressors for the current year reflect the following:-There is a direct correlation between the health expenditure in the budget year (HLTH_SHA11_HC (year)) and the social protection expenditure (SPR_REC_SUMT_ESC). This has been the case since 2010, when European policy underwent a paradigm shift from a conservative to a forward-looking orientation, focusing on disease prevention rather than treatment. This validates hypothesis H2: In the case of financial policy stability for health, the main deflator of the allocation efficiency is the social protection expenditure as a percentage of GDP (see Table 4). The approach is congruent with other papers [4,5,34].-There is a direct correlation between the expenditure allocated to health in the budget year (HLTH_SHA11_HC (year) and the social protection scheme approved at government level through national strategies and at European level through Community strategies (SPR_REC_SUMT_GOV (year)). This validates hypothesis H1: The health status of the population varies in direct proportion to the health budget allocations, and changes in the government policies are likely to have a direct proportional influence on the medium and long-term health balance (see Table 4). The same approach was used by [26,34].-There is a direct correlation between the expenditure allocated to health in the budget year (HLTH_SHA11_HC (year)) and the strategy to protect the health of the population in the long term (HLTH_SHA11_LT (year)), i.e., to populate health units with specialists and technical equipment to ensure the sustainability of the health services and the continuous increase in the level of health of the population.-The fluctuating dynamics of the Pearson correlation of the dependent variable in relation to social protection and health security expenditure suggests that the trend is polynomial, with the forecast line being affected (with an inflection point) during the pandemic period.-The dynamics of the social protection expenditure represented by administrative costs has a polynomial evolution on the Pearson correlation coefficients with the dependent variable defined by the equation:
y = −0.0771x + 1.3205(2)

Over the period 2013–2020, the Pearson coefficients of the financial indicator of social protection translated into administrative costs were aligned on a decreasing trend according to the regression equation:y = −0.0291x + 0.7958(3)

This shows that there is a 3% margin of reduction in financial allocations for social protection or for strengthening the health security of the population, which, in the context of the pandemic, has meant a pessimism expressed by the entire European population regarding vaccination. The intensity level of the pessimism is directly proportional to the level of allocations in each Member State for this type of expenditure (HLTH_SHA11_HC (year)). This development demonstrates hypothesis H4: Under the conditions of the health promotion through financial allocations for social protection or for strengthening the health security of the population, the vaccination rate tends to decrease if the funded level of the social security is not sustained over time. Our approach is also supported by [26,34].

All of the above aspects are highlighted in the Pearson correlation table of the dependent variable in relation to the regressors (see Table 4).

The results of this research reflect the fact that measures are needed to rebalance allocations in relation to health and social security policies, the recommended period of validity being 10 years, and that the pivots of expenditure should be refocused (in countries more affected by the pandemic, e.g., Romania) towards prevention measures and increasing health literacy among the population.

## 5. Conclusions

The present research aimed to assess the effectiveness of the health financial allocations from public funds within the EU.

As a result of the research, we found horizontal shortcomings (deviations among Member States from the public health policies promoted by the EU), but also vertical shortcomings, namely the destabilization of the allocation strategy during the COVID-19 pandemic.

We found that the change in the financial allocation paradigm from conservative to proactive had beneficial effects over the 10 years of implementation (2009–2018). Instead, the presence of the pandemic has captured underlying vulnerabilities in terms of the health culture of the population, the administrative capacity in the public health system of some Member States and the health status of the vulnerable pre-pandemic population in the face of these deficiencies.

Under these conditions, the measures adopted by the Member States during the pandemic have had as an effect the peaks of medical crisis (2020–2021), peaks in which morbidity and mortality related to SARS-CoV reached record levels in some Member States such as Italy, Spain, Sweden or Romania.

Post-marketing efforts to combat the disease have been slowed by a lack of health literacy among the population, amid insufficient financial support for this objective, which has reduced the impact of vaccination activities.

As measures, we have identified doing the following:-Rebalancing financial allocations for health in relation to strategic objectives at national and European levels;-Refocusing health police on prevention and health screening;-Linking short-term and long-term health policies;-Ensuring sustainability of the health sector through the sustainability of the resources used.

The novelty of this study lies in the retrospective and prospective approach to health financial projections in relation to health performance indicators and pandemic-related morbidity and mortality indicators. Moreover, the strategic directions for improvement correlated with the pattern of evolution of the efficiency of budget allocations in health can be a useful working tool for supranational decision-makers in order to readjust the optics of financing the sector and increasing the quality of health services.

The limitations of the study are the short time period considered for the analysis of SARS-CoV morbidity and mortality and the small number of indicators. The authors aim to address these shortcomings in future research.

## Figures and Tables

**Figure 1 ijerph-19-03063-f001:**
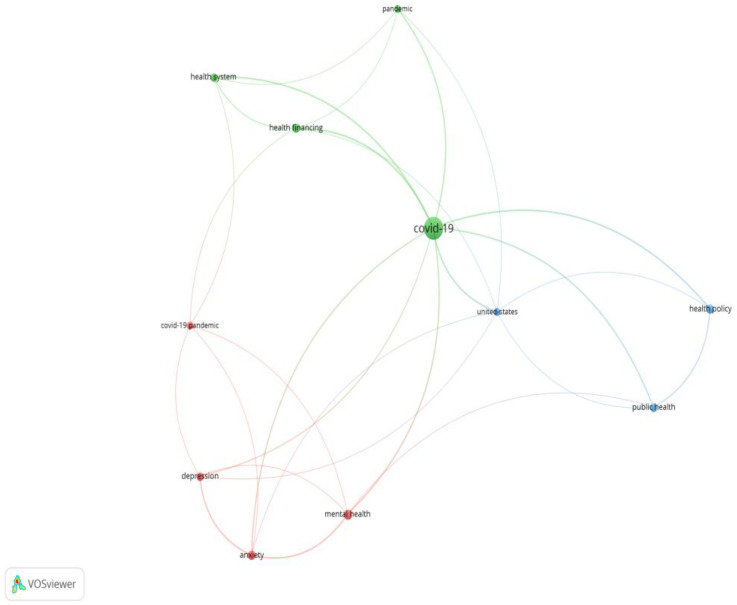
Cluster research publications regarding financing health services.

**Figure 2 ijerph-19-03063-f002:**
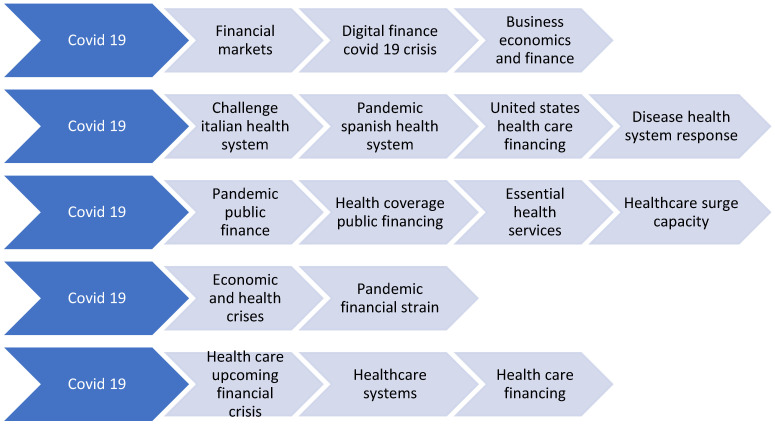
Research areas on health financing vs. pandemic.

**Table 1 ijerph-19-03063-t001:** Table of analysed indicators.

Indicators’ Names	Symbol	Unit of Measurement	Source
Expenditure on social protection	SPR_EXP_SUM	Percentage of GDP	[37]
Social benefits, in cash or in kind, to households and individuals	SPR_EXP_BNF	Percentage of GDP	[37]
Administration costs to the scheme for health management and administration	SPR_ADM	Percentage of GDP	[37]
Other expenditure by social protection schemes (payment of property income and other)	SPR_OTHER	Percentage of GDP	[37]
Receipts of social protection schemes (social contributions, general government contributions and other receipts)	SPR_REC_SUMT_GOV	Percentage of GDP	[37]
Employers’ social contributions	SPR_REC_SUMT_ESC-	Percentage of GDP	[37]
Total health care expenditure by EU27 average expenditure (Country average/EU27 average)	HLTH_SHA11_HC	Percentage of EU27 average expenditure	[37]
Long-term care (health) expenditure by EU27 average long-term expenditure (Country average/EU27 average)	HLTH_SHA11_LT	Percentage of EU27 average expenditure.	[37]
The rate of illness	COVID_ILL	Cases per 1 M people	[38]
The rate of vaccinations	COVID_VACC	% of population fully vaccinated	[38]

**Table 2 ijerph-19-03063-t002:** Annual model summary.

Model Year	R	R Square	Adjusted R Square	Std. Error of the Estimate	Change Statistics					Durbin-Watson
					R Square Change	F Change	df1	df2	Sig. F Change	
2009	0.999	0.998	0.996	2.365	0.998	734.070	10	16	0.000	2.059
2010	0.998	0.996	0.994	2.928	0.996	452.810	10	16	0.000	2.061
2011	0.998	0.996	0.994	3.013	0.996	443.429	10	16	0.000	2.131
2012	0.999	0.998	0.996	2.468	0.998	645.618	10	16	0.000	2.146
2013	0.999	0.998	0.997	2.075	0.998	863.058	10	16	0.000	2.303
2014	0.999	0.998	0.997	2.049	0.998	889.853	10	16	0.000	2.028
2015	1.000	0.999	0.999	1.253	0.999	2266.115	10	16	0.000	2.339
2016	0.999	0.999	0.998	1.569	0.999	1352.670	10	16	0.000	1.452
2017	0.999	0.999	0.998	1.454	0.999	1523.163	10	16	0.000	2.453
2018	1.000	0.999	0.998	1.365	0.999	1662.286	10	16	0.000	1.495
2019	0.999	0.999	0.998	1.423	0.999	1498.636	10	16	0.000	1.959
2020	0.998	0.996	0.994	2.649	0.996	443.106	10	16	0.000	1.910

**Table 3 ijerph-19-03063-t003:** ANOVA test.

Model Year	Sum of Squares Regression	Sum of Squares Residual	Sum of Squares Total	df Rg.	df Res	df Tot	Mean Square Regression	Mean Square Residual	F	Sig.
2009	41,072.167	89.522	41,161.689	10	16	26	4107.217	5.595	734.070	0.000
2010	38,825.678	137.190	38,962.868	10	16	26	3882.568	8.574	452.810	0.000
2011	40,249.734	145.231	40,394.965	10	16	26	4024.973	9.077	443.429	0.000
2012	39,328.383	97.465	39,425.848	10	16	26	3932.838	6.092	645.618	0.000
2013	37,149.809	68.871	37,218.680	10	16	26	3714.981	4.304	863.058	0.000
2014	37,373.159	67.199	37,440.357	10	16	26	3737.316	4.200	889.853	0.000
2015	35,586.873	25.126	35,611.999	10	16	26	3558.687	1.570	2266.115	0.000
2016	33,294.597	39.382	33,333.980	10	16	26	3329.460	2.461	1352.670	0.000
2017	32,190.421	33.814	32,224.236	10	16	26	3219.042	2.113	1523.163	0.000
2018	30,952.398	29.793	30,982.191	10	16	26	3095.240	1.862	1662.286	0.000
2019	30,344.785	32.397	30,377.182	10	16	26	3034.479	2.025	1498.636	0.000
2020	31,092.351	112.271	31,204.621	10	16	26	3109.235	7.017	443.106	0.000

**Table 4 ijerph-19-03063-t004:** Pearson Correlation Table between HLTH_SHA11_HC (year) and the other variables.

Year	HLTH_SHA11_HC (Year-1)	SPR_EXP_SUM (Year)	SPR_EXP_BNF (Year)	SPR_ADM (Year)	SPR_OTHER (Year)	SPR_REC_SUMT_ESC (Year)	SPR_REC_SUMT_GOV (Year)	HLTH_SHA11_LT (Year)	HLTH_SHA11_LT (Year-1)
2009	0.999	0.840	−0.546	0.581	0.272	0.306	0.580	0.854	0.850
2010	0.828	−0.497	0.572	0.180	0.308	0.566	0.998	0.864	0.868
2011	0.800	−0.397	0.558	0.038	0.376	0.574	0.987	0.856	0.896
2012	0.810	−0.405	0.610	−0.001	0.379	0.593	0.995	0.897	0.899
2013	0.797	−0.233	0.573	−0.090	0.410	0.558	0.998	0.895	0.899
2014	0.785	−0.428	0.578	0.054	0.403	0.559	0.998	0.898	0.903
2015	0.735	−0.439	0.546	0.097	0.346	0.568	0.999	0.902	0.907
2016	0.722	−0.461	0.587	0.100	0.288	0.582	0.999	0.900	0.905
2017	0.718	−0.493	0.576	0.158	0.277	0.598	0.999	0.904	0.911
2018	0.710	−0.632	0.576	0.318	0.334	0.597	0.999	0.912	0.917
2019	0.658	−0.551	0.570	0.187	0.334	0.574	0.999	0.915	0.918
2020	0.645	−0.527	0.569	0.149	0.314	0.560	0.996	0.919	0.905

## Data Availability

Not applicable.

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
