# Peer review of "Approaches to Health Efficiency across the European Space through the Lens of the Health Budget Effort"

_ijerph, 2022, doi:10.3390/ijerph19053063_

Round 1

Reviewer 1 Report

The paper ‘Approaches to Health Efficiency Across the European Space Through the Lens of The Health Budget Effort’ analyse a very interesting and actual topic, being based on appropriate econometric modelling, highlighting through relevant statistical indicators the intricacies of the existing relationship between the financial allocation to health and the expenditure on health protection of the population in the Member States.

We appreciate that, in general, the paper respects the scientific methodology and responds to the potential interest of readers on the topic, but it still needs the following minor improvements:

  1. In the Abstract and Conclusions, the impact and novelty of the research should be highlighted more.
  2. In the Introduction section, briefly outline the structure of the research.
  3. The Literature review section to be completed with more relevant papers in the field of research.

Author Response

Dear Reviewer 1,

Our research team have read your pertinent suggestions and made all necessary changes. Thank you for your support!

English language and style

( ) Extensive editing of English language and style required
( ) Moderate English changes required
( ) English language and style are fine/minor spell check required
(x) I don't feel qualified to judge about the English language and style

Yes

Can be improved

Must be improved

Not applicable

Does the introduction provide sufficient background and include all relevant references?

( )

(x)

( )

( )

Is the research design appropriate?

(x)

( )

( )

( )

Are the methods adequately described?

( )

(x)

( )

( )

Are the results clearly presented?

(x)

( )

( )

( )

Are the conclusions supported by the results?

(x)

( )

( )

( )

Comments and Suggestions for Authors

The paper ‘Approaches to Health Efficiency Across the European Space Through the Lens of The Health Budget Effort’ analyse a very interesting and actual topic, being based on appropriate econometric modelling, highlighting through relevant statistical indicators the intricacies of the existing relationship between the financial allocation to health and the expenditure on health protection of the population in the Member States.

We appreciate that, in general, the paper respects the scientific methodology and responds to the potential interest of readers on the topic, but it still needs the following minor improvements:

  1. In the Abstract and Conclusions, the impact and novelty of the research should be highlighted more.

Authors: We have supplemented the information on the novelty of the research with summary details in the Abstract and Conclusions, as recommended by you.

  1. In the Introduction section, briefly outline the structure of the research.

Authors: We have realized briefly outline according to your suggestions.

  1. The Literature review section to be completed with more relevant papers in the field of research.

Authors: We have supplemented the literature with a significant number of papers representative of the topic.

Reviewer 2 Report

Dear author(s)

The paper addresses a very interesting and actual topic. However, as to be considered for publication, substantial improvements should be made.

Literature Review

Page 2: please be more specific on the search terms on Google academic platform aiming to retrieve articles on the topic of healthcare financing.

Moreover, it is said that “Between 2000 and 2018, more than 70,000 articles on public health financing were written by international authors (see Capture 1).”. Since the keyword with the highest occurrence in capture 1 is COVID-19, which obviously was not a topic of interest in 200-2018 period and VoSViewer software cannot analyze data generated from Google Scholar, please review the content of the paragraph.

“A clustering of research interest has been identified in at least 11 clusters, ... “. In VosWiewer output, the nodes represent keywords, clustered by their cooccurrence in the same documents. The so formed clusters are represented by different color. Therefore, you have 11 keywords grouped in 3 clusters. (red, blue and green), for which you should provide meaningful explanation.

It is not clear how the content of figure 1 was drown from the bibliometric data.

Methodology

The research hypothesis are not part of the methodology. On the contrary, They should be formulated based on theoretical and empirical considerations, while the research methodology should be developed to support them (or not) by empirical data.

Please organize the research variables in a table, including their name, symbol, unit of measurement and source of each of them.

I suggest that equation 1 would be written by means of the equations word editor, since as they are now, The variables names are different from the ones above.

Please explain why for some of the variables was considered the previous year in the regression equation. Moreover, if available data for 10 consecutive years, why panel data regression was not employed in state of 10 yearly regressions.

Results and discussion

In terms of hypotheses testing, it is said that „The test results confirm hypothesis H3, namely: In the case of health financial policy stability, the pandemic morbidity rate tends to increase in magnitude, with the policy up-date level being a maximum of 10 years.” However, it is not clear which results supports the hypothesis, since in tables above there is no information on variables related to them. The same observation for the rest of hypotheses. If authors wish to validate the research hypotheses, there should presented meaningful research results.

Also, table 3 is named Pearson Correlation Table. However, Pearson Correlation coefficients should have values between -1 and 1. Moreover, in the first row of the first column is the MEAN word. This makes things more confusing.

Summarizing the above, please provide meaningful research results to support your hypotheses.

References

The theoretical background of the paper should be improved. Moreover, for a WoS indexed journal 22 references are not at all enough.

The references section and within text references should be formatted according to journal guidance.

Author Response

Dear Reviewer 2,

For the beginning, we want to thank you for your pertinent suggestions and support. We have read very carefully your observations and have made the necessary changes and  additions

Open Review

English language and style

( ) Extensive editing of English language and style required
( ) Moderate English changes required
( ) English language and style are fine/minor spell check required
(x) I don't feel qualified to judge about the English language and style

Yes

Can be improved

Must be improved

Not applicable

Does the introduction provide sufficient background and include all relevant references?

( )

( )

(x)

( )

Is the research design appropriate?

( )

( )

(x)

( )

Are the methods adequately described?

( )

( )

(x)

( )

Are the results clearly presented?

( )

( )

(x)

( )

Are the conclusions supported by the results?

( )

( )

(x)

( )

Comments and Suggestions for Authors

Dear author(s)

The paper addresses a very interesting and actual topic. However, as to be considered for publication, substantial improvements should be made.

Literature Review

Page 2: please be more specific on the search terms on Google academic platform aiming to retrieve articles on the topic of healthcare financing. Moreover, it is said that “Between 2000 and 2018, more than 70,000 articles on public health financing were written by international authors (see Capture 1).”. Since the keyword with the highest occurrence in capture 1 is COVID-19, which obviously was not a topic of interest in 2000-2018 period and VoSViewer software cannot analyze data generated from Google Scholar, please review the content of the paragraph.

Authors: The analysis covers the period 2000-2018 based on articles in

Google Scholar sub cheia de cautare: https://scholar.google.ro/scholar?q=finance+health+covid&hl=ro&as_sdt=0%2C5&as_vis=1&as_ylo=2000&as_yhi=2018. Please note that at the time of the research, 70500 articles were indexed for the period 2000-2018. For the same selection filter, but for the years 2019-2021, the research results were 45900, according to:  https://scholar.google.ro/scholar?q=finance+health+covid&hl=ro&as_sdt=0%2C5&as_vis=1&as_ylo=2019&as_yhi=2021.

If you wish, we can provide you with screenshots from that date (24.10.2021).

 “A clustering of research interest has been identified in at least 11 clusters, ... “. In VosWiewer output, the nodes represent keywords, clustered by their cooccurrence in the same documents. The so formed clusters are represented by different color. Therefore, you have 11 keywords grouped in 3 clusters. (red, blue and green), for which you should provide meaningful explanation.

Authors: We have corrected the number of clusters according to Capture 1 and realized meaningful explanation regarding Capture 1.

It is not clear how the content of figure 1 was drown from the bibliometric data.

Authors: In order to support the bibliometric analysis in the study, we have included the Appendix with the 75 targeted articles. On the other hand, we have supplemented the literature with a significant number of papers representative of the topic.

Methodology

The research hypothesis are not part of the methodology. On the contrary, They should be formulated based on theoretical and empirical considerations, while the research methodology should be developed to support them (or not) by empirical data.

Authors: We have moved the working hypotheses to the literature review section and linked them to the literature in the Results and discussion section.

Please organize the research variables in a table, including their name, symbol, unit of measurement and source of each of them.

Authors: We have inserted Table 1 according to your suggestions.

I suggest that equation 1 would be written by means of the equations word editor, since as they are now, The variables names are different from the ones above.

Authors: We have re-edited the equation for better visibility.

Please explain why for some of the variables was considered the previous year in the regression equation. Moreover, if available data for 10 consecutive years, why panel data regression was not employed in state of 10 yearly regressions.

Authors: Regression variables HLTH_SHA11_HC - Total health care expenditure by EU27 average expenditure and HLTH_SHA11_LT - Long-term care (health) expenditure by EU27 average long-term expenditure are quantified in the regression model for the previous years because we were entitled to highlight the dynamics of Pearson correlations for these indicators. Thus, at the level of 2009, the correlation of the dependent variable with the value of the previous year was 0.999, the trend being decreasing, as can be seen in the modified Pearson correlation table, since the table of mean dynamics was initially inserted. The other indicator reflects at the level of correlation with the dependent variable the dynamic correlation of the indicator for the current year and the previous year during the period analyzed, which by observation leads to an adequacy of the forecast as can be seen from the same Pearson correlation table.

Results and discussion

In terms of hypotheses testing, it is said that „The test results confirm hypothesis H3, namely: In the case of health financial policy stability, the pandemic morbidity rate tends to increase in magnitude, with the policy up-date level being a maximum of 10 years.” However, it is not clear which results supports the hypothesis, since in tables above there is no information on variables related to them. The same observation for the rest of hypotheses. If authors wish to validate the research hypotheses, there should presented meaningful research results.

Authors: We have made all the changes requested and made the necessary clarifications, which can be found in the text of the Results and discussions section.

Also, table 3 is named Pearson Correlation Table. However, Pearson Correlation coefficients should have values between -1 and 1. Moreover, in the first row of the first column is the MEAN word. This makes things more confusing.

Authors: We have replaced this table with the correct one on Pearson regressions.

Summarizing the above, please provide meaningful research results to support your hypotheses.

Authors: We realised it.

References

The theoretical background of the paper should be improved. Moreover, for a WoS indexed journal 22 references are not at all enough.

Authors: We have supplemented the literature with a significant number of papers representative of the topic.

The references section and within text references should be formatted according to journal guidance.

Authors: We did it.

Reviewer 3 Report

An interesting paper, but so simple and relating known issues. Anyway. literture review and introduction save this.

Author Response

Dear Reviewer 3,

We were shocked reading your opinion. As a result, we tried to improve our research significantly. We hope that this improved form of our paper will satisfy you.

Open Review

English language and style

( ) Extensive editing of English language and style required
( ) Moderate English changes required
( ) English language and style are fine/minor spell check required
(x) I don't feel qualified to judge about the English language and style

Yes

Can be improved

Must be improved

Not applicable

Does the introduction provide sufficient background and include all relevant references?

(x)

( )

( )

( )

Is the research design appropriate?

( )

(x)

( )

( )

Are the methods adequately described?

(x)

( )

( )

( )

Are the results clearly presented?

(x)

( )

( )

( )

Are the conclusions supported by the results?

(x)

( )

( )

( )

Comments and Suggestions for Authors

An interesting paper, but so simple and relating known issues. Anyway. literture review and introduction save this.

Authors: We have carried out a re-appraisal of the research on the basis of allegories relating to:

  • We have supplemented the information on the novelty of the research with summary details in the Abstract and Conclusions, as recommended by you.
  • We have realized briefly outline.
  • We have corrected the number of clusters according to Capture 1 and realized meaningful explanation regarding Capture 1.
  • In order to support the bibliometric analysis in the study, we have included the Appendix with the 75 targeted articles.
  • We have moved the working hypotheses to the literature review section and linked them to the literature in the Results and discussion section.
  • We have inserted a new Table 1.
  • We have re-edited the equation for better visibility.
  • We have significantly supplemented the literature with relevant reference articles.